# Impact of Autophagy Impairment on Experience- and Diet-Related Synaptic Plasticity

**DOI:** 10.3390/ijms23169228

**Published:** 2022-08-17

**Authors:** Ulyana Lalo, Ioannis P. Nezis, Yuriy Pankratov

**Affiliations:** School of Life Sciences, University of Warwick, Coventry CV4 7AL, UK

**Keywords:** autophagy, AMPA receptor, GABAA receptor, trafficking, LTP, synaptic scaling, synaptic current, Beclin1, endocytosis, VPS-34

## Abstract

The beneficial effects of diet and exercise on brain function are traditionally attributed to the enhancement of autophagy, which plays a key role in neuroprotection via the degradation of potentially harmful intracellular structures. The molecular machinery of autophagy has also been suggested to influence synaptic signaling via interaction with trafficking and endocytosis of synaptic vesicles and proteins. Still, the role of autophagy in the regulation of synaptic plasticity remains elusive, especially in the mammalian brain. We explored the impact of autophagy on synaptic transmission and homeostatic and acute synaptic plasticity using transgenic mice with induced deletion of the Beclin1 protein. We observed down-regulation of glutamatergic and up-regulation of GABAergic synaptic currents and impairment of long-term plasticity in the neocortex and hippocampus of Beclin1-deficient mice. Beclin1 deficiency also significantly reduced the effects of environmental enrichment, caloric restriction and its pharmacological mimetics (metformin and resveratrol) on synaptic transmission and plasticity. Taken together, our data strongly support the importance of autophagy in the regulation of excitatory and inhibitory synaptic transmission and synaptic plasticity in the neocortex and hippocampus. Our results also strongly suggest that the positive modulatory actions of metformin and resveratrol in acute and homeostatic synaptic plasticity, and therefore their beneficial effects on brain function, occur via the modulation of autophagy.

## 1. Introduction

An active lifestyle and physical exercise are widely believed to benefit both physical and mental health [1,2]. There have been numerous reports of the positive effects of an active lifestyle and its proxy in animal experiments—environmental enrichment—on cognitive function, both in human patients and rodent models [2,3,4,5]. The cognitive effects of physical exercise (PE) and environmental enrichment (EE) rely on a specific form of synaptic plasticity—homeostatic synaptic scaling—an ability of synapses to change their efficacy in response to the activity of neuronal networks [6,7,8]. The remarkable responsiveness of synapses to various experiences, such as environmental enrichment, physical activity or changes in diet, can underlie perspective therapeutic approaches to ameliorate the progression of age-related brain disorders [9,10,11].

Apart from physical exercise, caloric restriction (CR), usually defined as a reduced intake of calories not causing malnutrition, is widely suggested as an efficient strategy to improve health- and lifespan in most living organisms [3,9,12,13]. Over the years, numerous studies have reported that CR can exert a variety of life-extending effects and counteract various age-associated biochemical alterations, both in experimental models and human patients [1,3,9,12,13,14].

It is commonly accepted that the beneficial effects of CR originate mainly from the enhancement of autophagy [3,10,15,16]. Autophagy (more specifically, macroautophagy) is a complex cellular process in which misfolded or aggregated proteins and damaged organelles are engulfed within double-membrane vesicles (autophagosomes) and delivered to lysosomes for degradation [17,18]. As a catabolic process, autophagy is very sensitive to decreases in nutrients levels, so it could be readily activated by CR and fasting regimens. Indeed, CR was widely reported to induce or enhance autophagy via several pathways, including the activation of AMP-activated kinase (AMPK), which in turn inhibits the autophagy suppressor mTOR (mechanistic target of rapamycin) and the regulation of sirtuins, causing protein deacetylation and enhancement of autophagy [9,13,19].

Conversely, pharmacological modulators of AMPK, mTOR and sirtuins have been shown to reproduce many of the autophagy-related effects of CR; such drugs are widely referred as “CR mimetics” [9,13,16]. Since usage of long-term CR protocols can be challenging for many elderly human patients (e.g., due to such comorbidities as diabetes or vascular dementia), interest in the usage of autophagy modulators mimicking the effect of CR on health-span is rapidly growing [12,13,15,18,20]. Furthermore, CR and CR mimetics are widely used as part of an ample approach to induce autophagy in different types of experimental animals [12,13].

Regulation of autophagy in mammalian cells critically depends on the activity of the Beclin1–vacuolar sorting protein 34 complex (Beclin1-VPS34 complex) [12,14,17,18,21]. The Beclin1-VPS34 complex works downstream of the main pathways of autophagy initiation, including the nutrient and energy sensors mTOR and AMPK and deacetylases (e.g., sirtuins) [9,14,15]. Thus, transgenic animals with inducible impairment of Beclin1 can serve as very useful models to test the impact of autophagy impairment on neuronal function. Such a model of inducible genetic deletion of Beclin1 in Purkinje neurons was used to explore the role of the VPS34-mediated pathway in neuron viability [22].

Although the beneficial effects of autophagy on brain function have been traditionally attributed to the digestion of potentially harmful intracellular structures, e.g., protein aggregates [3,14,18], there is growing evidence that the molecular machinery of autophagy can also directly affect synaptic dynamics and plasticity, putatively via interaction with the presynaptic mechanisms of vesicular trafficking and postsynaptic cascades of endocytosis and degradation of neurotransmitter receptors and ion channels [21,23,24]. Still, most of the studies reporting various molecular interactions between autophagy-related proteins and elements of synaptic machinery have been carried out in non-mammalian models [14,18,24], and the role of autophagy in the regulation of synaptic plasticity in the mammalian brain remains elusive. For instance, both decrease and increase in autophagy have been linked (in different studies) to the enhancement of synaptic plasticity in different pathological contexts [13,14,18,23,24]. Correspondingly, the key role for autophagy in the beneficial effects of caloric restriction diet and active lifestyle on synaptic transmission and cognitive function has yet to be fully established.

In this work, we explored the effect of autophagy impairment on synaptic transmission and long-term synaptic plasticity in cortical neurons using transgenic mice with induced deficiency of the Beclin1 protein. We also tested whether Beclin1 deficiency alters the effects of environmental enrichment and caloric restriction and its pharmacological mimetics on synaptic function.

## 2. Results

### 2.1. Impairment of Autophagy in Becn1-Deficient Neocortical and Hippocampal Neurons

Since full *Becn1* deletion can cause embryonic lethality and thereby prevent physiological studies in adult brain tissue [22], we used transgenic mice with inducible partial Beclin1 deficiency. To generate a conditional knock-out mouse line (hereafter referred to as Beclin1-KO or *Becn1*^+/−^), we first crossed *Gt(ROSA)26S-CAG*-*GCaMP6f* ^Fl/Fl^ mice with *Becn1*^Fl/Fl^ mice to generate a *GCaMP6f* ^Fl/Fl^ × *Becn1*^Fl/Fl^ line and then crossed the progeny with Camk2a^creERT2/+^ transgenic mice (see Methods) to induce the Becn1 deficiency in the pyramidal neurons of the hippocampus and neocortex, which abundantly express CaMKIIa. As a control line (hereafter referred to as WT or *Becn1*^+/+^ mice), we used *GCaMP6f* ^Fl/Fl^ × *Becn1*^+/+^ × Camk2a^creERT2/+^ mice (Figure 1). Expression of fluorescent reporter protein GCaMP6f in the targeted neurons facilitated their identification in vitro and in situ (Figure 1A); it also allowed us to assess their vitality by observation of evoked Ca^2+^ signaling (Figure 1B).

The partial Becnlin1 deficiency did not have detrimental effects on the vitality of the neocortical and hippocampal neurons and did not lead to significant neuronal loss, as evidenced by the lack of significant differences in the cross-immunostaining against neuronal marker NeuN and GCaMP6 in the *Becn1*^+/−^ mice (Figure 1A). We also observed robust Ca^2+^-transients in the pyramidal neurons of the *Becn1*^+/−^ mice (Figure 1B), both in the hippocampal CA1 and neocortical L2/3 areas. Although we observed a statistically significant difference in the amplitude of evoked Ca^2+^-transients in the *Becn1*^+/−^ mice and their *Becn1*^+/+^ littermates (Figure 1B), this difference was not dramatic and could be explained by the difference in glutamatergic transmission (as described below). There were also only subtle differences in the basic electrophysiological properties of neocortical pyramidal neurons, such as the input resistance and amplitude of voltage-gated Na currents (Figure 1C). We would like to emphasize that, by any means, these differences cannot explain the differences in the amplitudes of synaptic currents described below.

At the same time, *Becn1*^+/−^ mice exhibited marked impairment of autophagy. We assessed the autophagic flux in live neocortical and hippocampal neurons by changes in staining with the specific lysosomal marker *LysoView633* after incubation with the lysosomal inhibitor Bafilomycin A1 [25] (Figure 2A,B). The relative increase in the BafilomycinA1-induced punctate *LysoView633* staining was much lower in the neurons of *Becn1*^+/−^ mice as compared to their WT counterparts. Furthermore, exposure to CR led to much higher increases in the autophagic flux in neurons of *Becn1*^+/+^ mice (Figure 2B,C). Similarly, 3 h-long incubation of brain slices with the CR mimetics metformin (5 μM) and resveratrol (20 μM) significantly augmented the effect of BafilomycinA1 in the WT but not in the Beclin1-KO mice (Figure 2D). Of note, metformin and resveratrol at the concentrations used in our experiments have been reported to enhance autophagy [20,26]; also, these concentrations lay within the range achievable in preclinical studies in human patients [9,20,26].

The above data demonstrate that inducible Beclin1 deficiency caused an impairment of autophagy in hippocampal and neocortical neurons without serious detrimental effects on their vitality. Our data also verify the positive modulatory effects of CR, metformin and resveratrol on neuronal autophagy in the adult mammalian brain.

### 2.2. Impact of Beclin1 Deficiency on Inhibitory Synaptic Transmission

To investigate the role of autophagy in the mechanisms of experience-dependent homeostatic synaptic plasticity, we examined the effect of CR and EE on basal synaptic transmission in neocortical pyramidal neurons. One of the most likely candidate mechanisms might be autophagy-mediated modulation of GABA_A_ receptor trafficking via the Beclin1-VSP34-GABARAP cascade, which has been implicated in the effects of the VSP34 inhibitor SAR405 on inhibitory neurotransmission in the amygdala [27].

We used whole-cell voltage-clamp recordings to register GABA_A_ receptor-mediated spontaneous miniature inhibitory postsynaptic currents (mIPSCs) at a membrane potential of −80 mV in the presence of tetrodotoxin (TTX, 1 μM) and antagonists of AMPA (DNQX, 30 µM) and P2X (PPADS, 10 µM) receptors (Figure 3A,B). First of all, the average amplitude of GABAergic mIPSCs recorded in the neocortical layer 2/3 neurons of Becn1-KO mice kept under standard housing (SH) conditions (27.8 ± 4.9 pA, n = 10) was significantly larger than in the WT SH mice (20.2 ± 5.8 pA, n = 14). The analysis of the amplitude distributions of mIPSCs (see Methods) showed a similar difference in their quantal amplitude (or unitary response) (Figure 3C,D) in the Becn1-KO vs. WT mice (correspondingly, 21.8 ± 4.6 pA and 16.6 ± 4.3 pA). Similar to the neocortical neurons, the mIPSCs recorded in the CA1 pyramidal neurons exhibited up-regulation in the Becn1-KO mice (Figure 3D). In parallel, there was no significant difference in the average frequency of basal mIPSCs in the Becn1-KO and WT mice, both in the neocortical and hippocampal neurons (Figure 3D). These results strongly suggest the postsynaptic locus of the effect of Becn1 deficiency on inhibitory synaptic currents.

Exposure of *Becn1*^+/+^ mice to CR led to significant decrease in the quantal size and average amplitude of mIPSCs (Figure 3A,C,D) as opposed to the *Becn1*^+/−^ mice, in which the CR-induced decrease of inhibitory currents was not statistically significant (*p* > 0.3 at a statistical power of 0.9). The CR-induced reduction in mIPSCs in the CA1 pyramidal neurons was also Becn1-dependent (Figure 4D).

At the same time, environmental enrichment caused profound decrease in the inhibitory currents in the neocortical neurons of WT mice and moderate yet statistically significant decrease in the Becn1-KO mice (Figure 3A,C). Both after CR and EE, decrease in mean mIPSC amplitude was accompanied by a similar decrease in quantal size, whereas the frequencies of spontaneous events did not undergo significant changes (Figure 3D). These data strongly support the key role of Becn1 and CR-induced autophagy enhancement in the postsynaptic regulation of GABAergic inhibitory synaptic transmission. The effect of EE on GABAergic synapses also turned out, rather surprisingly, to be partially dependent on Becn1 (Figure 3D), since the quantal size of neocortical mIPSCs recorded in the EE-exposed *Becn1*^+/−^ mice was moderately lower (18.8 ± 5.3 pA, n = 7) than in the SH counterparts. The difference between the EE and SH *Becn1*^+/−^ mice was statistically significant (*p* < 0.05).

### 2.3. Impact of Beclin1 Deficiency on Excitatory Synaptic Transmission

The impact of autophagy on excitatory synaptic signaling remains uncertain, especially at the postsynaptic locus [18,23]. To explore the effects of Beclin1 deficiency on glutamatergic excitatory transmission, we recorded the AMPA receptor-mediated spontaneous miniature excitatory synaptic currents (mEPSCs) in the neocortical and hippocampal pyramidal neurons in the presence of TTX, the GABA_A_ receptor antagonist picrotoxin (100 µM) and PPADS (10 µM). We observed a marked deficit in glutamatergic synaptic transmission in the neurons of *Becn1*^+/−^ mice, which manifested in a significant decrease in the mean amplitude and quantal size of mEPSCs (Figure 4). The quantal size of mEPSCs recorded in the neocortical neurons of WT and Becn1-KO mice in standard housing (Figure 4C,D) was 12.2 ± 3.8 pA (n = 13) and 7.4 ± 2.7 pA (n = 11), respectively. Exposure to CR and EE led to the significant up-regulation of glutamatergic synaptic signaling in the WT mice (Figure 4A,D); the quantal size of mEPSCs recorded in the neocortical neurons of CR- and EE-treated WT mice increased to 17.1 ± 4.6 pA (n = 10) and 15.6 ± 4.7 pA (n = 9), respectively. The CR-induced up-scaling of mEPSCs was abolished in the *Becn1*^+/−^ mice, whereas exposure to EE produced a moderate but statistically significant effect (Figure 4C,D). The quantal amplitude of mEPSCs measured in the EE-treated *Becn1*^+/−^ mice was higher (9.5 ± 2.6 pA) that in the SH *Becn1*^+/−^ mice. The excitatory synaptic currents recorded in the CA1 pyramidal neurons exhibited a similar pattern of changes in the Beclin1-deficient mice (Figure 4D), namely, a decrease in the baseline amplitude and lack of CR-induced up-scaling.

Interestingly, the postsynaptic changes in the amplitude of glutamatergic currents in the *Becn1*^+/−^ mice were accompanied by notable changes in the frequency of spontaneous events, both in the neocortex and hippocampus (Figure 4D). The frequency of mEPSCs registered in the layer 2/3 pyramidal neurons of SH *Becn1*^+/−^ mice was 0.75 ± 0.36 Hz as compared to 1.21 ± 0.44 Hz (*p* < 0.02) in the SH WT mice. Exposure to CR and EE did not cause significant changes in mEPSC frequency in the mice of both genotypes. It is worth noting that decrease in mEPSC frequency in the neurons of *Becn1*^+/−^ mice cannot be attributed to reduction in their amplitude below the threshold of detection (typically, 2.5–2.8 pA), since the amplitude of spontaneous currents registered under all conditions was higher than the amplitude of the background noise (as indicated in Figure 4C). The above results strongly suggested that the changes in mEPSC frequency in the *Becn1*^+/−^ mice had, most likely, a presynaptic origin.

Taken together, our data implicate autophagy and the Beclin1-mediated pathway, in particular, in the regulation of glutamatergic signaling in the neocortical excitatory synapses both at the post- and presynaptic *loci*.

### 2.4. Impact of CR Mimetics on Synaptic Transmission in the Neocortex

To study the effects of the CR mimetics metformin and resveratrol on neuronal signaling in situ over a wide time range, we combined two experimental approaches: (i) we monitored parameters of inhibitory or excitatory synaptic currents in one and the same neuron during application of these drugs for up to 60 min, in which time periods stable whole-cell recordings (variation in the input resistance and average mEPSC amplitude and frequency less than 15%) were achieved in 90% of the neurons tested; and (ii) we compared the parameters of synaptic currents recorded under control conditions with recordings obtained of the neocortical neurons of brain slices preincubated with drugs for 3–6 h. While the former approach would allow detection of the acute effects of the drugs, the latter would enable us to explore more long-term action. In all cases, the number of neurons taken for the analysis and report was nine, both for *Becn1*^+/+^ and *Becn1*^+/−^ mice.

Since autophagy-related effects of CR mimetics would, most likely, be linked to modulation of trafficking or endocytosis of postsynaptic proteins and synaptic vesicles, one could hardly expect them to develop within the first 10–20 min after drug application. Therefore, the presence of acute effects would suggest the direct modulatory action of the drugs on postsynaptic receptors and/or release of neurotransmitters rather than autophagy-related mechanisms; this could be verified further by comparing the effects in the *Becn1*^+/+^ and *Becn1*^+/−^ mice.

The bath application of metformin (5 µM) to the neocortical neurons of *Becn1*^+/+^ mice in situ did not produce any notable acute effect on GABAergic mIPSCs within the first 30 min (Figure 5A,C), but at the time point of 60 min it caused a significant (up to 25%) decrease in the average amplitude of mIPSCs. The metformin-induced decrease in the mIPSC amplitude was even larger (13.8 ± 3.6 vs. 20.4 ± 5.7 pA in the control) in the neurons of brain slices kept with the drug for 3 h; increase in the incubation time did not reduce the amplitude any further. There were no significant changes in the mIPSC frequencies at any time point, suggesting the postsynaptic locus for the effect of metformin.

The time course of the effect of resveratrol (20 µM) on mIPSCs in the neurons of *Becn1*^+/+^ mice was similar to that of metformin, showing a marked reduction in mIPSC amplitude after 1 h of application (Figure 5B,C). The magnitude of the resveratrol-induced reduction in the inhibitory current (27% of control) was slightly lower than the effect of metformin (34% of control).

Thus, long-term incubation of brain slices of wild-type mice with resveratrol and metformin had rather similar effects (reductions in amplitude) on inhibitory synaptic currents as the exposure of animals to CR. Importantly, the effects of both resveratrol and metformin were dramatically diminished in the neurons of *Becn1*^+/−^ mice, strongly supporting the involvement of Beclin1-regulated autophagy in the effects of these CR mimetics. Still, it is worth noting that the effect produced by metformin at 6 h was not completely abolished in the Beclin1-KO mice, which might imply the involvement of some other molecular cascades.

Using a similar experimental paradigm, we explored the action of metformin and resveratrol on glutamatergic excitatory synaptic currents (Figure 6). Similar to the mIPSCs, neither drug produced any acute effects on the mEPSCs. The 3 h-long incubation of the wild-type neocortical neurons with resveratrol caused a significant increase in mEPSC amplitude (18.8 ± 3.5 vs. 13.6 ± 4.1 pA in the control), whereas metformin caused only moderate (12–18% of control) up-regulation, which was statistically significant only at the 6 h time point (Figure 6A–C). Neither drug had a significant effect on the frequencies of the mEPSCs.

Again, the action of CR mimetics on glutamatergic transmission in the Beclin1-deficient mice differed considerably to that in their wild-type littermates (Figure 6A–C). The effect of the 6 h-long incubation with metformin on mEPSC amplitude was abolished in the *Becn1*^+/−^ mice. Interestingly, the average amplitude of mEPSCs in the neurons of Beclin1-deficient mice was even smaller (by 9% compared to the control) after the long-term incubation with resveratrol. These results agree with the notion of resveratrol and metformin acting as autophagy-dependent CR mimetics.

### 2.5. Impact of Beclin1 Deficiency on Synaptic Plasticity in the Neocortex and Hippocampus

The results described above (Figure 3 and Figure 4) suggest that the strength of excitatory synapses can be down-regulated in the Beclin1-deficient mice, whereas inhibitory synaptic transmission can be up-regulated. Potentially, these alterations could have a negative effect on long-term synaptic potentiation. On the other hand, the EE- and CR-induced up-scaling of glutamatergic synapses (Figure 3), in synergy with down-regulation of GABAergic transmission (Figure 5), could have a positive modulatory effect on long-term synaptic plasticity, which could also be reproduced by CR mimetics. To test these hypotheses, we investigated the long-term potentiation (LTP) of the field excitatory postsynaptic potentials (fEPSPs) in the layer 2/3 neurons of the somatosensory cortex and the CA1 hippocampal areas of the wild-type and Beclin1-KO mice (Figure 7 and Figure 8). 

In the neocortex, we used two protocols of theta-burst stimulation to induce long-term changes in synaptic transmission (as described previously in [28,29,30,31]; see Methods, also). In the wild-type mice under control conditions, the strong stimulus consisting of five episodes of theta-burst high-frequency stimulation (5 TBS) induced a robust increase in the fEPSP slope (142 ± 17% of control) in all 15 experiments (Figure 6A). At the same time, the usage of a weaker stimulation consisting of two theta-burst episodes (2 TBS) induced a mild long-term depression in fEPSPs (92 ± 12%, n = 11). As previously discussed in [31,32], such sharp stimulus-dependence of LTP induction makes it an effective tool for exploring the mechanisms of synaptic homeostasis and metaplasticity. In the neocortical neurons of Beclin1-deficient mice, the stimulus-dependence of the neocortical LTP was different to that in their wild-type counterparts (Figure 7A,C): the weaker stimulation did not induce marked changes in the fEPSPs (Figure 7B), while the magnitude of LTP induced by the strong stimulation (117 ± 14%, n = 12) was significantly lower than in the *Becn1*^+/+^ mice (Figure 7C).

Exposure of the WT mice to EE led to a moderate increase in the fEPSP potentiation induced by 5 TBS (155 ± 19% of control, n = 9) but caused dramatic changes in the effect of the weaker stimulation, which induced a rather strong, long-term increase (125 ± 12%, n = 7) in the fEPSP slope in the EE-exposed mice. Exposure of the WT mice to the caloric restriction diet also had a strong effect on the LTP (Figure 7C). Neither EE nor CR caused any significant effect on the LTP induced by the strong stimulus in the neocortex of Beclin1-deficient mice (Figure 6). In contrast, the “switching” effect of EE and CR on the long-term plasticity induced by weaker stimulation was retained in the *Becn1*^+/−^ mice; the long-term increases in the fEPSP slope reached 119 ± 11% (n = 10) and 123 ± 13% (n = 12), respectively. Still, in both cases, there was a statistically significant difference in the magnitude of 2 TBS-induced LTP in the *Becn1*^+/+^ and *Becn1*^+/−^ mice.

Preincubation of brain slices of WT mice with metformin for 4–6 h reproduced the effect of CR on the neocortical LTP for both weaker and stronger stimulation strengths. The incubation with resveratrol had an even stronger effect on 5 TBS-induced LTP, enhancing its magnitude by up to 171 ± 22% of the baseline (n = 9). The facilitatory action of both drugs was significantly reduced in the neocortex of *Becn1*^+/−^ mice. The sensitivity of metformin action to Beclin1 deficiency appeared to be higher (no statistically significant enhancement in *Becn1*^+/−^ mice) than that of resveratrol, which still caused a statistically significant effect in the *Becn1*^+/−^ mice (Figure 7C).

To evaluate synaptic plasticity in the CA1 hippocampal area, we used a conventional approach (see Methods), in which fEPSPs are elicited by the stimulation of *Schaffer collaterals* and LTP is induced by one theta-burst of high frequency stimulation (Figure 8). As in the neocortex, the LTP magnitude was significantly reduced in the CA hippocampal neurons of Beclin1-deficient mice (Figure 8A). Since LTP is NMDA receptor-dependent both in the neocortical and pyramidal neurons and therefore requires strong post-tetanic depolarization, one might explain the deficit in the LTP observed in the *Becn1*^+/−^ mice (Figure 7A and Figure 8A) simply by the up-regulation of GABAergic inhibition. To test this hypothesis, we induced LTP in the CA1 area in the presence of a moderate concentration (1 µM) of GABA_A_ receptor inhibitor gabazine. At such concentrations, gabazine inhibited the GABAergic synaptic currents in the layer 2/3 and CA1 neurons by 26 ± 8 (n = 5) and 29 ± 7% (n = 4, data not shown), respectively, and thereby might counterbalance the effect of Beclin1 deficiency. However, the attenuation of GABAergic inhibition by gabazine did not rescue the LTP in the hippocampus (Figure 8A,C) and the neocortex (n = 5; data not shown) of the *Becn1*^+/−^ mice, which implies that impairment of LTP in these mice has a more complex mechanism than simple up-regulation of inhibitory synaptic transmission.

The magnitude of the CA1 LTP underwent a moderate but statistically significant increase both in the wild-type and the Beclin1-deficient mice exposed to EE (Figure 8C). In contrast, exposure to CR had a significant effect on the LTP only in the wild-type mice (Figure 8C). The effect of CR on hippocampal LTP was fully reproduced in the brain slices of the *Becn1*^+/+^ mice preincubated with resveratrol (Figure 8B,C). In the *Becn1*^+/−^ mice, resveratrol caused a moderate but statistically significant increase in the LTP magnitude (169 ± 18% vs. 142 ± 17% in the control). The preincubation of brain slices with metformin caused very strong enhancement of the CA1 LTP in the wild-type mice (Figure 8B); the effect of metformin was abolished in the Beclin1-deficient mice (Figure 8B,C). The weaker dependence on *Becn1* deletion (in comparison to metformin and CR) of resveratrol’s action on LTP, observed both in the neocortex (Figure 7) and hippocampus (Figure 8), suggests a contribution of some autophagy-independent molecular cascades. 

Taken together, our data strongly support the importance of Beclin1-mediated autophagy in the regulation of excitatory and inhibitory synaptic transmission and synaptic plasticity in the neocortex and hippocampus. Our results also verify that the positive modulatory actions of metformin and resveratrol in acute and homeostatic synaptic plasticity, and therefore their beneficial effects on brain function, can occur mainly via the modulation of autophagy.

## 3. Discussion

There is an emerging consensus on the importance of the modulation of autophagy for neuronal function and homeostasis [5,12,23,33]. Traditionally, the beneficial effects of autophagy, particularly in the context of caloric restriction, have been attributed to improved neuroprotection, whereas negative consequences of autophagy impairment have mainly been associated with impaired proteostasis and decreased vitality of neurons [12,13,14]. Despite the growing number of reports of molecular interactions between autophagy-related proteins and synaptic machinery, predominantly in non-mammalian models [14,18,24], observations of the direct effects of the modulation of autophagy on synaptic plasticity in mammalian brain tissue remain scarce [18,23,33].

In this work, we demonstrate that autophagy deficiency can exert direct effects on synaptic function. Our data show alterations in the efficacy of glutamatergic synapses in a novel murine model of impaired autophagy. Furthermore, our results highlight the importance of autophagy for the modulation of synaptic transmission at postsynaptic loci and imply autophagy-related cascades in the mechanisms of homeostatic synaptic plasticity.

In our first line of experiments (Figure 1 and Figure 2), we verified that inducible Beclin1 deficiency did not exert severe effects on basic electrophysiological characteristics and neuronal vitality but led to a decrease in autophagic flux; it also prevented the effects of metformin and resveratrol—the well-established modulators of autophagy. Thus, the *Becn1*^+/−^ mice can be considered as an adequate model of mild deficit in autophagy, which can occur, for instance, during the early stages of age-related neurodegenerative diseases. At the same time, the bulk of our data on synaptic functions (Figure 3, Figure 4, Figure 5, Figure 6, Figure 7 and Figure 8) revealed considerable differences in the synaptic signaling and plasticity in the *Becn1*^+/−^ mice as compared to their wild-type littermates. These observations strongly support the notion that most important molecular and functional changes in neurons, which underlie cognitive decline, can occur at the earlier stages of neurodegenerative diseases, well before the accumulation of toxic aggregates and massive neuronal loss [18,34].

Interestingly, our results revealed two distinct trends in the alterations in baseline synaptic transmission in the Becline1-deficient mice: increase in the postsynaptic efficacy (i.e., quantal size) of inhibitory synapses without marked changes in the frequency of synaptic events (Figure 3) contrasts with decrease both in the quantal amplitude and in the frequency of excitatory synaptic currents (Figure 4). Our observations of up-regulation of GABAergic signaling in the neocortical and hippocampal neurons of Beclin1-deficient mice at postsynaptic loci (Figure 3) closely agree with the effects of the selective autophagy inhibitor SAR405 on inhibitory transmission in the amygdala [27] and strongly support the instrumental role of autophagy in the regulation of inhibitory synaptic transmission by modulation of GABA_A_ receptor trafficking via the Beclin1-VSP34-GABARAP cascade [21,22,27,33]. The pattern of changes in excitatory synaptic transmission in the Beclin1-deficient mice was more complex and intriguing. Firstly, the decrease in the quantal amplitude of AMPA receptor-mediated synaptic currents (i.e., decrease in postsynaptic efficacy) in mice with inhibited neuronal autophagy was rather unexpected, since previous reports suggested that degradation of postsynaptic AMPA receptors and decrease in the strength of excitatory synapses can be caused by the up-regulation of autophagy [23,35,36,37]. Secondly, we observed a certain negative effect of Beclin1 deficiency at presynaptic loci, namely, a decrease in the frequency of spontaneous glutamatergic mEPSCs. Such presynaptic down-regulation of the excitatory synapses in the autophagy-deficient mice do not entirely agree with the notion that autophagy can slow down the kinetics of synaptic vesicle recycling and decrease their readily releasable pool, presumably via directing them to autophagosomes [18,24]. So, the presynaptic mechanisms of autophagy-related modulation of synaptic efficacy require further investigation.

On the other hand, the decrease in the frequency of glutamatergic events in the autophagy-deficient mice could have had a postsynaptic component, for instance, the degradation of synaptic spines. Combined with the leftward shift of the amplitude distribution of AMPA receptor-mediated mEPSCs (Figure 4), this hypothesis closely agrees with data that suggest an instrumental role for autophagy in the maintenance and rejuvenation of synaptic densities and active zones [12,15,23]. We would also like to note that our results on the negative effect of autophagy impairment on glutamatergic synaptic transmission and the data of other groups on the negative effects of autophagy induction on synaptic strength [35,36,37] were obtained in different neurons and under different experimental paradigms. Taken together, these data highlight the crucial importance of autophagy for the homeostatic regulation of glutamatergic synapses, the molecular mechanisms and the (patho)physiological implications of which are yet to be explored, especially in mammalian neurons. One of the most interesting candidate mechanisms is the impact of autophagy on endocytosis and the lysosomal sorting of glutamate receptors, which can exert bidirectional effects on the strength of excitatory synapses [23,38,39,40].

Both down-regulation of excitatory and up-regulation of inhibitory synaptic currents in the *Becn1*^+/−^ mice are consistent with similar trends for age-related changes in synaptic transmission which we reported previously [30,32]. These age-related changes in synaptic signaling were ameliorated by caloric restriction and environmental enrichment [30], which did not have substantial effects on the baseline mEPSCs and mIPSCs in the Beclin1-deficient mice (Figure 3 and Figure 4). Collectively, these results implicate the regulation of autophagy in mechanisms of homeostatic synaptic plasticity and imply that even a moderate deficiency in autophagy can contribute to age-related changes in synaptic transmission and thereby to cognitive decline.

Our observations of LTP impairment in the neocortex (Figure 7) and hippocampus (Figure 8) of the *Becn1*^+/−^ mice are in line with emerging evidence of links between autophagy and long-term synaptic plasticity and memory [27,35,40,41,42]. The most parsimonious explanation of the decrease in the LTP magnitude in the Beclin1-deficient mice observed in the present work might be the synergetic effect of changes in baseline synaptic transmission: decrease in the excitatory synaptic input and increase in GABAergic inhibition can reduce the post-tetanic depolarization required for efficient activation of NMDA receptors, thereby hampering LTP induction. However, our result showing that attenuation of inhibitory synaptic input with moderate concentrations of GABA_A_ receptor antagonists was not able to rescue LTP in CA1 neurons of *Becn1*^+/−^ mice (Figure 8B) strongly counts against such a simple explanation and suggests the involvement of more complex mechanisms. Indeed, there is a growing number of reports implicating the autophagic degradation of many key synaptic proteins, such as NMDA and AMPA receptors and synaptic scaffolding proteins, in the plasticity of excitatory synapses [18,24,35,37,38,40,41]. Still, both increase and decrease in autophagy have been reported to cause impairment of different types of synaptic plasticity and memory [14,18,23,24,42], so the molecular mechanisms underlying the role for autophagy in learning and memory remain elusive.

The present study also provided new insights related to the involvement of Beclin1-mediated autophagy in the effects of exercise, caloric restriction and their putative pharmacological mimetics metformin and resveratrol on synaptic signaling. Although an instrumental role for autophagy in the beneficial effects of these factors on brain health has been widely assumed [1,3,5,11,13,18,19], this notion has not been verified in mammalian models of impaired autophagy. In general, our data confirm the “CR-mimicking” action of metformin and resveratrol (Figure 5, Figure 6, Figure 7 and Figure 8). We showed that modulatory effects of CR, metformin and resveratrol on excitatory and inhibitory synaptic transmission and their facilitatory action on LTP in the neocortex and hippocampus were strongly reduced in the Becn1-deficient mice, which strongly supports the pivotal role for autophagy in these effects. However, the effects of resveratrol on synaptic plasticity in the neocortical and hippocampal neurons were only partially sensitive to the *Becn1* deletion (Figure 7C and Figure 8B,C), suggesting the participation of some autophagy-independent mechanisms. One such mechanism might be the enhancement of astroglial Ca^2+^ signaling and glia–neuron interactions, reported previously [32,43].

Interestingly, the modulatory action of EE on synaptic transmission and plasticity turned out to be dependent on autophagy as well (Figure 3, Figure 4, Figure 7 and Figure 8), though it was not completely abolished in the *Becn1*^+/−^ mice (Figure 3D, Figure 4D and Figure 8C). This result implies the involvement of other mechanisms in the effects of EE on synaptic dynamics, most likely via molecular cascades triggered by BDNF [30,35,44], which can be intensively released during physical exercise (an important component of EE) [1,45].

To conclude, our results strongly suggest that even mild deficiency in autophagy, which does not seriously compromise the vitality of neurons and does not lead to significant synaptic loss, can cause opposing changes in the efficacy of glutamatergic and GABAergic synapses and thereby significantly affect synaptic plasticity in the neocortex and hippocampus. On the other hand, our data show that exogenous modulators of autophagy, such as metformin and resveratrol, can mimic the positive effects of diet and exercise on synaptic dynamics and plasticity.

## 4. Materials and Methods

### 4.1. Experimental Animals

All animal work was carried out in accordance with UK legislation and the “3R” strategy; the research did not involve non-human primates. This project was approved by the University of Warwick Animal Welfare and Ethical Review Body (AWERB) (approval number: G13–19) and regulated under the auspices of the UK Home Office Animals (Scientific Procedures) Act licenses P1D8E11D6 and I3EBF4DB9.

To generate the mice with conditional deletion of Becn1 gene in cortical neurons, floxed mutant mice possessing loxP sites flanking exon 2 of the Becn1 gene [22] were crossed with transgenic mice expressing the tamoxifen-inducible Cre recombinase under the control of the mouse Camk2a (calcium/calmodulin-dependent protein kinase II alpha) promoter region (Camk2a^creERT2/+^ mice, JAX stock number 012362). The Becn1^Flox/Flox^ mice, generated as described in [22], were obtained from the Jackson Laboratory (JAX Stock number 028794). To provide co-expression of fluorescent reporter protein, the Becn1^Flox/Flox^ mice were crossed with Gt(ROSA)26S-CAG-*GCaMP6f* ^Flox/Flox^ mice (JAX stock 029626). The double-transgene *GCaMP6f* ^Flox/Flox^xBecn1^Flox/Flox^ offspring were crossed with Camk2a^creERT2/+^ transgenic mice to induce the Becn1 deficiency (referred to as *Becn1*^+/−^ or Beclin1-KO mice) in the pyramidal neurons of the hippocampus and neocortex. To create the control mouse line (referred to as WT or *Becn1*^+/+^ mice), the *Becn1*^+/+^ mice were crossed with *GCaMP6f* ^Flox/Flox^ and then with Camk2a^creERT2/+^ mice. The genotypes of all animals were verified by PCR using ear samples. 

Gene expression was induced in the mice at 8–12 weeks of age with tamoxifen (150 mg/kg, once per day for 3 days). The mice were used for the experiments at age 12–20 weeks, 4–8 weeks after the tamoxifen injection. After the injection, the mice were kept either under standard housing conditions (SH) or exposed to the enriched environment (EE), including ad libitum access to a running wheel, or kept on a mild calorie-restriction diet for 4–6 weeks [30,44]. According to the protocol for caloric restriction, food intake was reduced by 10–20% and was individually regulated to maintain body weight loss of 10–15%.

### 4.2. Electrophysiological Recordings

The mice were anaesthetized with halothane, then decapitated; the brains were removed rapidly after decapitation and placed into ice-cold physiological saline containing (mM): NaCl 130, KCl 3, CaCl_2_ 0.5, MgCl_2_ 2.5, NaH_2_PO_4_ 1, NaHCO_3_ 25 and glucose 15, with a pH of 7.4 and gassed with 95% O_2_–5% CO_2_. Transverse slices (260 μm) were cut at 4 °C and then placed in physiological saline containing (mM): NaCl 130, KCl 3, CaCl_2_ 2.5, MgCl_2_ 1, NaH_2_PO_4_ 1, NaHCO_3_ 22 and glucose 15, with a pH of 7.4 and gassed with 95% O_2_–5% CO_2_, and kept for 1–5 h prior to imaging and electrophysiological recording.

Whole-cell voltage-clamp recordings from neocortical and hippocampal neurons were performed using glass patch-pipettes (4–5 MΩ) filled with intracellular solution (in mM): 60 CsGluconate, 50 CsCl, 10 NaCl, 10 HEPES, 5 MgATP, 1 D-Serine, 0.1 EGTA, pH 7.35. Transmembrane currents were monitored using a MultiClamp 700B patch-clamp amplifier (Axon Instruments, Austin, TX, USA), filtered at 2 kHz and digitized at 4 kHz. Experiments were controlled with a Digidata1440A data acquisition board (Axon Instruments, Austin, TX, USA) and WinWCP software (Strathclyde University, Edinburgh, UK); data were analyzed using self-designed software. Liquid junction potentials were compensated with the patch-clamp amplifier. The series and input resistances were, respectively, 5–7 MΩ and 800–1200 MΩ; both series and input resistance varied by less than 15% in the cells accepted for analysis. When spontaneous synaptic currents were recorded, the inhibitor of P2X purinoreceptors PPADS (10 µM) was added to the extracellular solution to prevent possible interference from the small fraction of purinergic excitatory spontaneous currents [31,46,47] on the amplitude distribution of mEPSCs.

In the synaptic plasticity experiments, the same slice preparations were used. Field excitatory postsynaptic potentials (fEPSPs) were measured using a glass micropipette filled with extracellular solution (0.5–1 MΩ resistance) placed in neocortical layer 2/3 neurons or in the stratum radiatum of the CA1 area. The fEPSPs were evoked by the stimulation of neuronal afferents descending from layers 4–5 or the stimulation of CA3-CA1 Schaffer collaterals with a bipolar coaxial electrode (WPI, Stevenage, UK); stimulus duration was 300 µs, and the stimulus strength was set to provide an average fEPSP amplitude of about 30–40% of the maximal level (typically, 1.5–3 μA in the neocortex and 1.2–2.5 μA in the hippocampus). LTP was induced by 1 (CA1 neurons), 2 (L2/3 neurons) or 5 (L2/3 neurons) episodes of theta-burst stimulation (TBS), each TBS episode consisting of 10 pulses of 100 Hz stimulation, repeated 10 times with a 200 ms interval (a total of 100 pulses per episode).

### 4.3. Multi-Photon Fluorescent Imaging 

Two-photon images of neurons were acquired at a 5Hz frame rate using a Zeiss LSM-7MP multi-photon microscope coupled to a SpectraPhysics MaiTai pulsing laser; the experiments were controlled by ZEN LSM software (Carl Zeiss, Jena, Germany). Images were further analyzed offline using ZEN LSM (Carl Zeiss) and ImageJ (NIH) software. The changes in the concentration of cytoplasmic free Ca^2+^ ([Ca^2+^]_i_) in situ were assessed by changes in *GCaMP6f* fluorescence. The [Ca^2+^]_i_ levels were expressed as ΔF/F ratio averaged over a region of interest (ROI). To quantify the net response of a cell to stimulation, a Ca^2+^ signal was integrated over 3 min immediately after agonist application, averaged over the whole-cell image and normalized to the baseline integral Ca^2+^ signal.

Autophagy was assessed in live pyramidal neurons of the somatosensory cortex and hippocampus in situ by staining with highly specific pH-sensitive lysosomal marker *LysoView633* (Molecular Probes, Austin, TX, USA). The 10 min-long incubation with 100 nM *LysoView633* was followed by a 15 min-long washout and was repeated twice: in the control and after 40 min incubation with the lysosomal inhibitor Bafilomycin A1 (1 µM). Autophagic flux was quantified by the relative increase in the percentage of cell area (exhibiting green *GCamp6* fluorescence) occupied by the *LysoView*-labeled puncta. For the immunolabelling for neuronal markers and GCaMP6f, neocortical and hippocampal neurons were incubated with mouse anti-NeuN (1:750; Abcam, Cambridge, UK) and chicken anti-GFP (1:750; Abcam ab4674) antibodies, as described elsewhere [28,44,46,48]. Prior to cell loading, antibodies were conjugated to the fluorescent dyes Texas Red (NeuN) and fluorescein using the Lighting-Link antibody conjugation system (Innova Bioscience, Cambridge, UK), according to the manufacturer’s protocol. Fluorescence was excited at 820 nm; GCamp6 and fluorescein signals were observed at 520 ± 10 nm, TexasRed was observed at 590 ± 20 nm, and the *LysoView633* Texas Red signal was observed at 650 ± 20 nm.

### 4.4. Drugs

Receptor antagonists were from Tocris-BioTechne (Abingdon, UK). Other salts and chemicals were from Sigma-Merck (Dorset, UK), unless otherwise indicated.

### 4.5. Data Analysis 

All data are presented as means ± standard deviations (SDs) and the statistical significance of differences between data groups was tested by two-tailed unpaired *t*-tests, unless otherwise indicated. For all cases of statistical significance reported, the statistical power of the test was 0.8–0.9. Each brain slice was used for only one experiment (e.g., fluorescent/electrophysiological recordings in a single neuron or single LTP experiment). The number of experiments/cells reported is therefore equal to the number of slices used. The experimental protocols were allocated randomly, so the data in any group were drawn from at least 3 animals, typically from 4 to 8 mice. The average ratio of experimental units per animal was 1.31 for the LTP experiments, 1.48 for the whole-cell recordings and fluorescent Ca^2+^-measurements, and 1 for the immunohistochemistry. The spontaneous transmembrane currents recorded in neurons were analyzed off-line using methods described previously [28,46,47,49]. The amplitude distributions of synaptic currents were analyzed with the aid of probability density functions and likelihood-maximization techniques; all histograms shown were calculated as probability density functions. The amplitude distributions were fitted with either multi-quantal binomial models or bi-modal functions consisting of two Gaussians with variable peak location, width and amplitude; the parameters of models were fitted using a likelihood-maximization routine, as described previously [47,49].

## 5. Conclusions

Our results strongly support the importance of autophagy in the regulation of excitatory and inhibitory synaptic transmission and synaptic plasticity in the neocortex and hippocampus. Our data show that deficiency in autophagy can cause opposing changes in the efficacy of glutamatergic and GABAergic synapses, which, in turn, can significantly affect long-term synaptic plasticity in the neocortex and hippocampus. Our data also show that exogenous modulators of autophagy, such as metformin and resveratrol, can mimic the positive effects of diet and exercise on synaptic dynamics and plasticity.

## Figures and Tables

**Figure 1 ijms-23-09228-f001:**
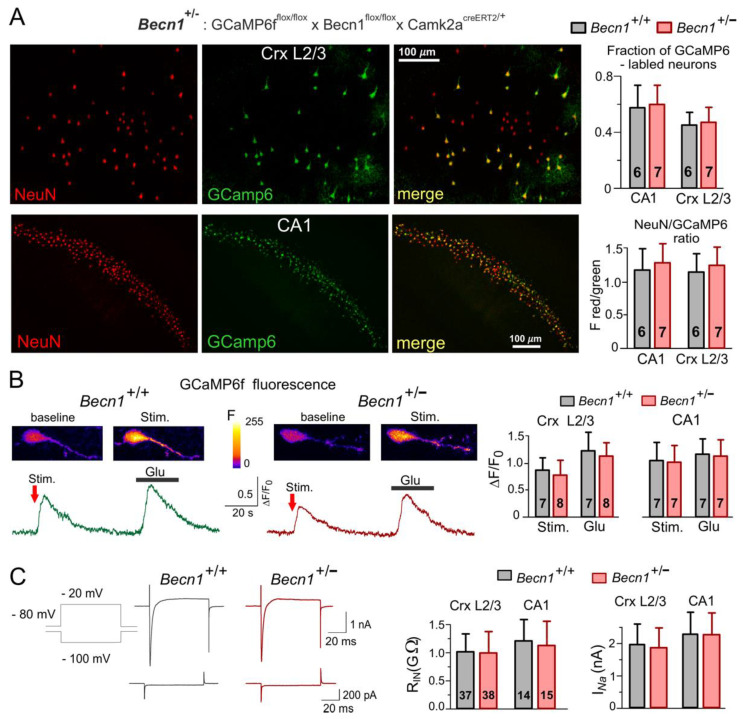
Impairment of autophagy in the Beclin1-deficient mice did not severely affect the basic functions of neocortical and hippocampal neurons. Mice were obtained by crossing *GCaMP6f*^fl/fl^ × *Becn1*^fl/fl^ × *Camk2a*^creERT2/+^ (*Becn1*^+/−^) and GCaMP6f^fl/^ × Camk2a^creERT2/+^ (*Becn1*^+/+^) lines. (**A**) The GCaMP6f-expressing cells were immunostained against neuronal marker NeuN. *Left*: representative two-photon fluorescent images of neocortical CR mimetics/3 and the CA1 hippocampal area. *Right*: data on the fraction of NeuN-labelled cells also expressing GCaMP6f, measured in the CA1 region and somatosensory cortex CR mimetics/3 and pooled for the number of mice indicated (three to four slices for each animal). (**B**) The cytosolic Ca^2+^-transients evoked in the GCaMP6-expressing neocortical pyramidal neurons by synaptic stimulation and application of glutamate (20 µM). *Left* panels show representative pseudo-colour multi-photon fluorescent images recorded at rest and at the peak of Ca^2+^-transients shown below. *Right* panel shows the pooled data (means ± SDs for the numbers of neurons indicated) of peak Ca^2+^ response in L2/3 and CA1 neurons; the difference between *Becn1*^+/+^ and *Becn1*^+/−^ mice was not statistically significant. (**C**) Basic electrophysiological characterization of neocortical and hippocampal pyramidal neurons. *Left* panel shows the representative whole-cell voltage-clamp currents elicited in the neocortical neuron by the change in holding membrane potential from −80 mV to −100 mV (lower waveforms) and to −20 mV (upper waveforms). Note the high input resistance and the presence of fast voltage-gated Na^+^ currents in both the *Becn1*^+/+^ and *Becn1*^+/−^ mice. *Right* panel shows the pooled data on the input resistance (R_In_) and amplitude of Na^+^ currents (I_Na_) in layer 2/3 and CA1 neurons (means ± SDs for the numbers of neurons indicated). Note the lack of difference in the basic functional properties of neurons in the wild-type and Beclin1-deficient mice.

**Figure 2 ijms-23-09228-f002:**
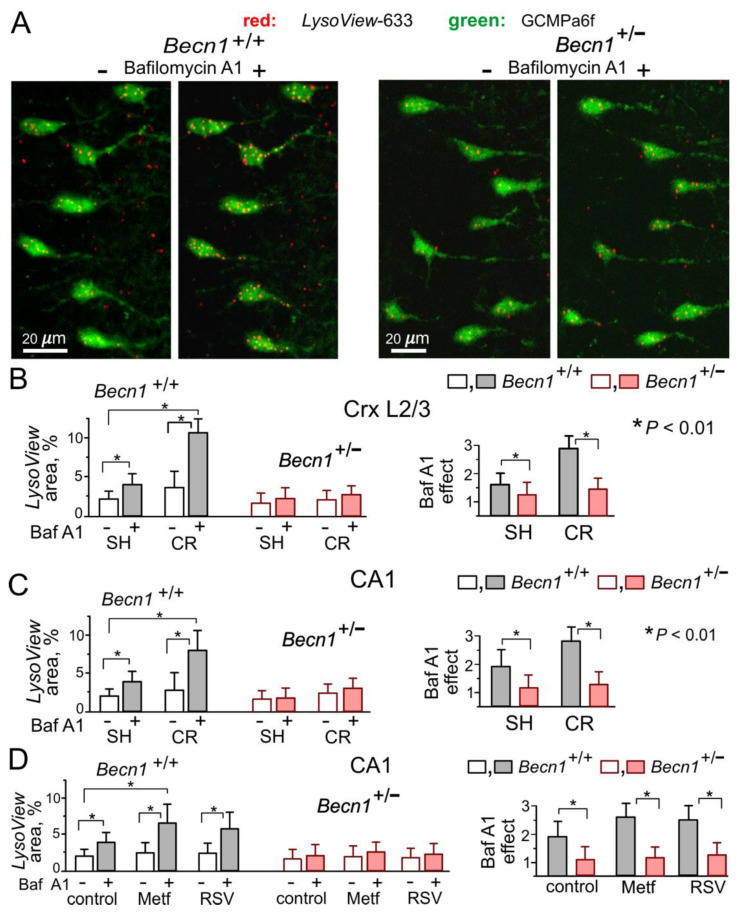
Impairment of autophagy in the neurons of Beclin1-deficient mice. Autophagy was assessed in live pyramidal neurons of the somatosensory cortex and hippocampus by staining with the highly specific pH-sensitive lysosomal marker *LysoView633* in the control and after a 40 min-long incubation with the lysosomal inhibitor Bafilomycin A1 (1 μM), as described in the *Methods*. Autophagic flux was quantified by the increase in the fraction of cell area (green GCamp6-fluorescence) occupied by the LysoView-labeled puncta (red fluorescence). (**A**) The representative two-photon images of layer 2/3 neocortical neurons in *Becn1*^+/−^ and *Becn1*^+/+^ mice before and after incubation with Bafilomycin A1. (**B**,**C**) The pooled data (means ± SDs for 40–50 cells from 3 mice) on the punctate *LysoView* staining (left) and the effect of Bafilomycin A1 (right) in layer 2/3 and CA1 neurons, assessed in the mice in standard housing and exposed to CR. (**D**) The pooled data on the autophagic flux in CA1 pyramidal neurons, assessed in the mice in standard housing in the control and in the presence of metformin (Meft) and resveratrol (RSV). Note the significant CR-induced enhancement of autophagic flux in the *Becn1*^+/+^ but not the *Becn1*^+/−^ mice, and the difference in effects of metformin and resveratrol in the *Becn1*^+/+^ and *Becn1*^+/−^ mice. * *p* < 0.01.

**Figure 3 ijms-23-09228-f003:**
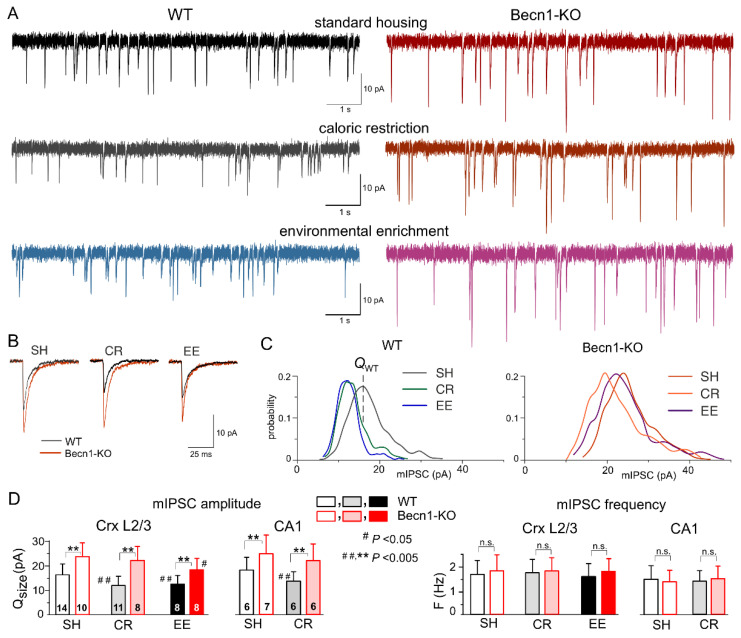
Impact of autophagy impairment on baseline inhibitory synaptic transmission and homeostatic synaptic plasticity. GABA_A_ receptor-mediated miniature inhibitory synaptic currents (mIPSCs) were recorded in the neocortical and hippocampal pyramidal neurons at −80 mV in the presence of 30 µM DNQX, 1 µM TTX and 10 µM PPADS. (**A**) The representative whole-cell currents recorded in the neocortical layer 2/3 neurons of the *Becn1*^+/+^ (left column) and *Becn1*^+/−^ (right column) mice in standard housing (top row) and exposed to CR (middle row) and EE (bottom row). (**B**) The average mIPSCs waveforms recorded under different conditions as indicated (average of 20 mIPSCs each). Note the significant increase in the mIPSC amplitude in the Becnlin1-deficient mice. (**C**) The corresponding amplitude distributions (probability density functions calculated for 300–600 events). The quantal size of mIPSCs was evaluated by the peak of amplitude distribution (as indicated for the neuron of the WT SH mouse) and then confirmed by maximum likelihood analysis (see Methods). The postsynaptic decrease in the quantal size of mIPSCs in WT mice exposed to CR and EE was evidenced by the leftward shift of the peak of amplitude distribution. (**D**) Diagrams show the quantal size (evaluated as shown in panel **C**) and frequency of the mIPSCs recorded in the neocortical layer 2/3 and CA1 neurons of WT and Beclin1-deificient mice of different treatment groups; data are shown as the means ± SDs for the numbers of neurons indicated. The asterisks (**) correspondingly indicate the statistical significance of the differences between the *Becn1*^+/+^ and *Becn1*^+/−^ mice. The hash symbols (# or ##) indicate the statistical significance of the effects of CR or EE as compared to the SH mice of the same genotype. Note the significant increase in the quantal amplitude of mIPSCs and the lack of change in their frequencies in the Beclin1-deficient mice.

**Figure 4 ijms-23-09228-f004:**
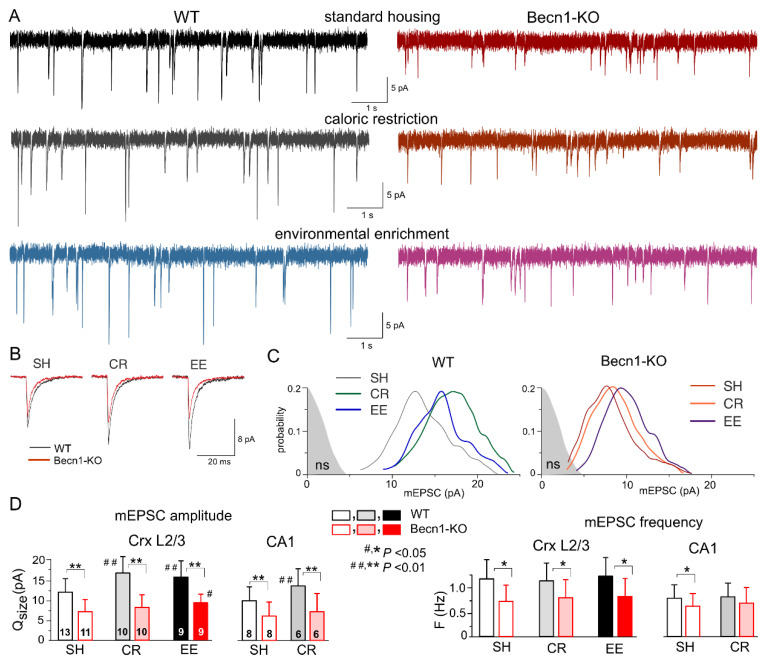
Impact of autophagy impairment on baseline excitatory synaptic transmission and homeostatic synaptic plasticity. The AMPA receptor-mediated miniature excitatory synaptic currents (mEPSCs) were recorded in the neocortical and hippocampal pyramidal neurons at −80 mV in the presence of 100 µM picrotoxin, 1 µM TTX and 10 µM PPADS. (**A**) The representative whole-cell currents recorded in the neocortical layer 2/3 neurons of the *Becn1*^+/+^ (left column) and *Becn1*^+/−^ (right column) mice in standard housing (top row) and exposed to CR (middle row) and EE (bottom row). (**B**) The average mEPSC waveforms recorded under different conditions, as indicated (average of 20 mEPSCs each). Note the significant decrease in the mEPSC amplitude in the Becnlin1-deficient mice. (**C**) The corresponding amplitude distributions (probability density functions calculated for 300–600 events); the shaded areas show the amplitude distribution of background noise. The postsynaptic decrease in the quantal size of mEPSCs in the Beclin1-deficient mice (as compared to WT) was evidenced by the leftward shift of the peak of amplitude distribution. Note that the amplitudes of most of the mEPSCs in the neuron of Beclin1-deficient are still above the noise. (**D**) Diagrams show the quantal size (evaluated as shown in panel **C**) and frequency of the mEPSCs recorded in the layer 2/3 and CA1 pyramidal neurons of WT and Beclin1-deficient mice of different environment groups; data are shown as means ± SDs for the numbers of neurons indicated. The asterisks (* or **) correspondingly indicate the statistical significance of the difference between the *Becn1*^+/+^ and *Becn1*^+/−^ mice. The hash symbols (# or ##) indicate the statistical significance of the effects of CR or EE as compared to the SH mice of the same genotype. Note the significant decrease in the quantal amplitude and frequency of mEPSCs in the Beclin1-deficient mice.

**Figure 5 ijms-23-09228-f005:**
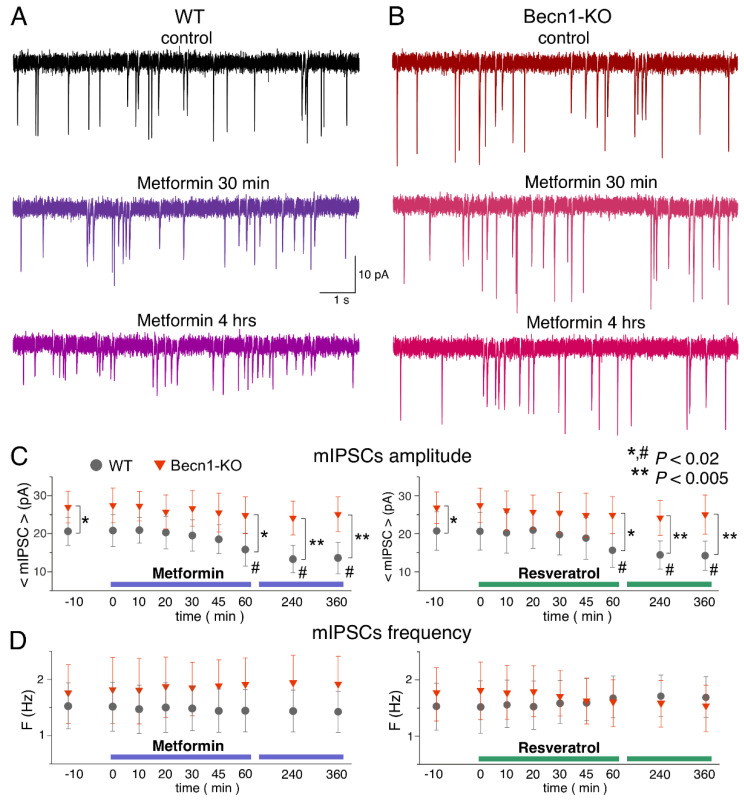
Effects of autophagy modulators on inhibitory synaptic transmission in the neocortical neurons. The GABAergic mIPSCs were recorded as in the Figure 3. (**A**,**B**) The representative mIPSCs recorded in the neurons of the *Becn1*^+/+^ (**A**) and *Becn1*^+/−^ (**B**) mice in the control (top row) and after application of metformin (5 µM) for 30 min (middle row, the same neuron as in the control) and 4 h (bottom row); the scale is the same in all panels. Note the significant decrease in the mIPSC amplitude in the WT neuron after 4 h-long application of metformin. (**C**,**D**) Each dot shows the average amplitude and frequency of spontaneous currents recorded in a 2 min time window in the pyramidal neurons of wild-type and Beclin1-deficient mice during application of metformin (5 µM) and resveratrol (20 µM). The drugs were applied at the zero time point; data are presented as means ± SDs for nine neurons. The asterisks (* or **) indicate the statistical significance of the differences between the *Becn1*^+/+^ and *Becn1*^+/−^ mice. The hash symbols (#) indicate the statistical significance of the effects of drugs as compared to the neuron of the same genotype in the control. A decrease in the amplitude of mIPSCs was observed in the wild-type but not in the Beclin1-deficient mice.

**Figure 6 ijms-23-09228-f006:**
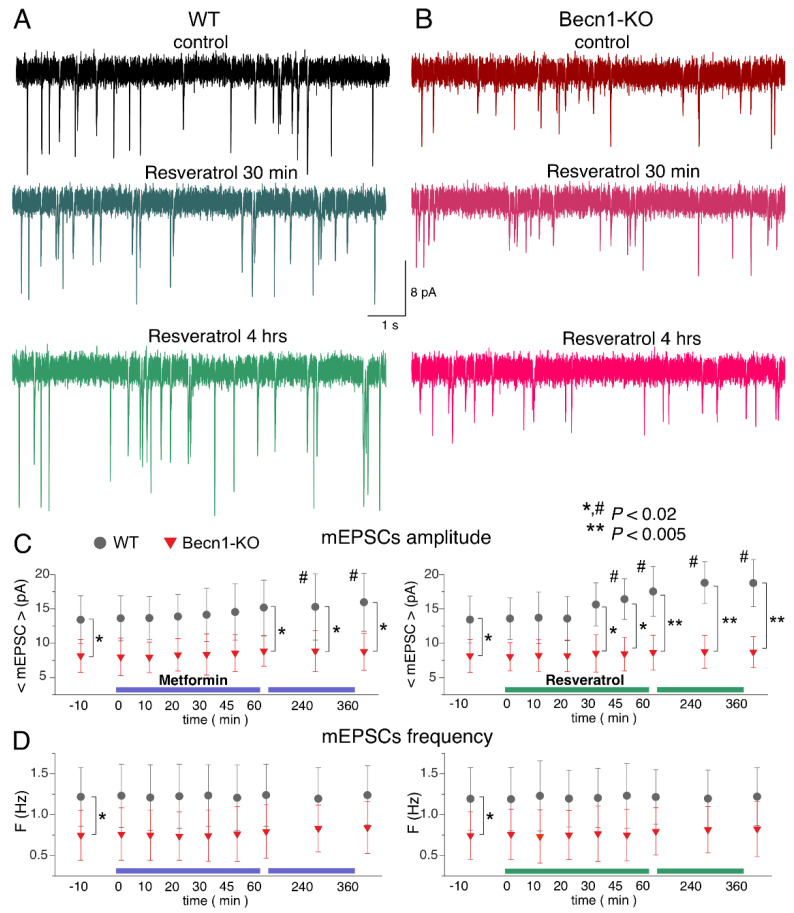
Effects of autophagy modulators on excitatory synaptic transmission in the neocortical neurons. The glutamatergic mEPSCs were recorded as in Figure 3. (**A**,**B**) The representative mEPSCs recorded in the neurons of the *Becn1*^+/+^ (**A**) and *Becn1*^+/−^ (**B**) mice in the control (top row) and after application of metformin (5 µM) for 30 min (middle row, the same neuron as in the control) and 4 h (bottom row); the same scale applies for all panels. Note the significant increase in the mEPSC amplitude in the WT neuron after 4 h-long application of metformin. (**C**,**D**) Each dot shows the average amplitude and frequency of spontaneous currents recorded in a 2 min time window in the pyramidal neurons of the wild-type and Beclin1-deficient mice during the application of metformin (5 µM) and resveratrol (20 µM). The drugs were applied at the zero time point; data are presented as means ± SDs for nine neurons. The asterisks (* or **) indicate the statistical significance of the differences between the *Becn1*^+/+^ and *Becn1*^+/−^ mice. The hash symbols (#) indicate the statistical significance of the effects of the drugs as compared to the neuron of the same genotype in the control. Decrease in the amplitude of mEPSCs was observed in the wild-type but not in the Beclin1-deficient mice.

**Figure 7 ijms-23-09228-f007:**
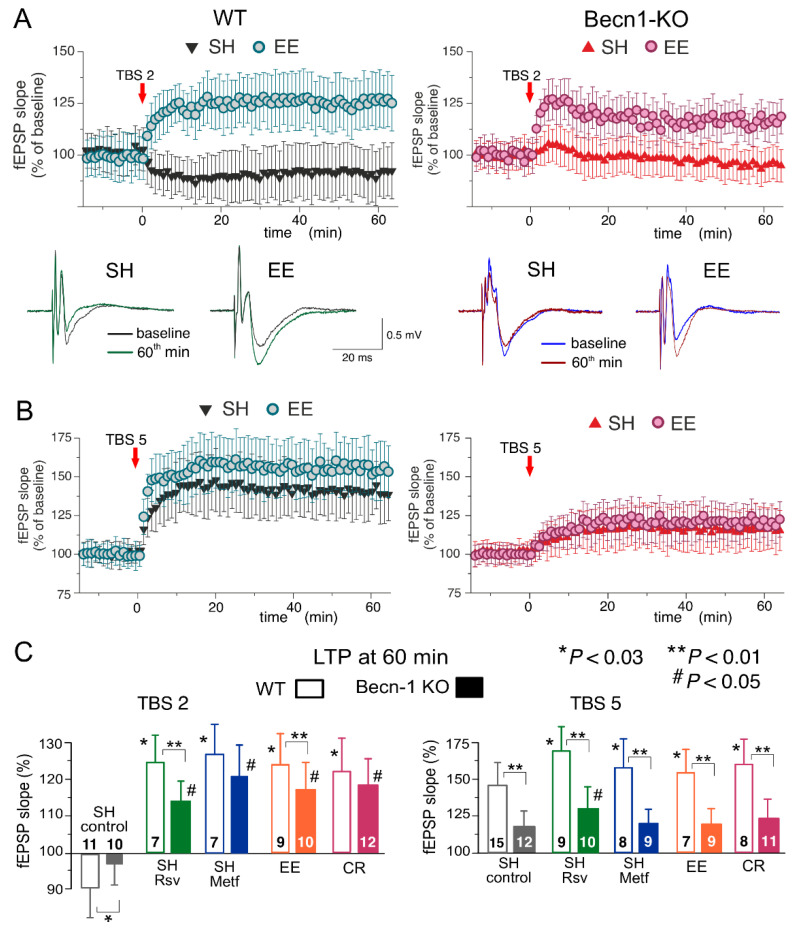
Impact of autophagy impairment on long-term synaptic plasticity in the neocortex. The LTP of the field EPSPs was induced in the layer 2/3 neurons of the somatosensory cortex of the wild-type and Beclin1-deficient mice by two (**A**) or five (**B**) episodes of theta-burst stimulation (TBS), as described in *Methods*. (**A**,**B**) The time course of changes in the slopes of fEPSPs recorded in the mice in standard housing (SH) and mice exposed to the enriched environment (EE). Dots in the graphs represent the average of six consecutive fEPSPs; data are shown as means ± SDs for 7–15 experiments (as indicated in panel **C**). Data were normalized to the fEPSP slope averaged over the 10 min period prior to the TBS. The insets show the representative average fEPSP waveforms recorded in the SH and EE mice of corresponding genotype before and 60 min after the TBS. (**C**) The pooled data on the magnitude of the LTP in the neocortex of mice of both genotypes at different conditions. In the SH mice, the LTP was evaluated in the control or after preincubation of brain slices with metformin (Metf) or resveratrol (RSV) for >4 h, as shown in Figure 5 and Figure 6. Graphs show the magnitude of LTP evaluated as relative increase in the fEPSP slope at 60 min, averaged across a 10 min time window. Data are shown as means ± SDs for the numbers of experiments indicated. Statistical significance (two-population unpaired *t*-test) for the differences between genotypes and treatment groups is indicated by different symbols, as follows: (*) effect of EE, CR or drug in the WT mice as compared to the control; (#) effect of EE, CR or drug in the Beclin1-deifient mice as compared to the control; (**) difference between the WT and Beclin1-deficient mice. Note the significant decrease in the LTP magnitude in Beclin1-KO mice.

**Figure 8 ijms-23-09228-f008:**
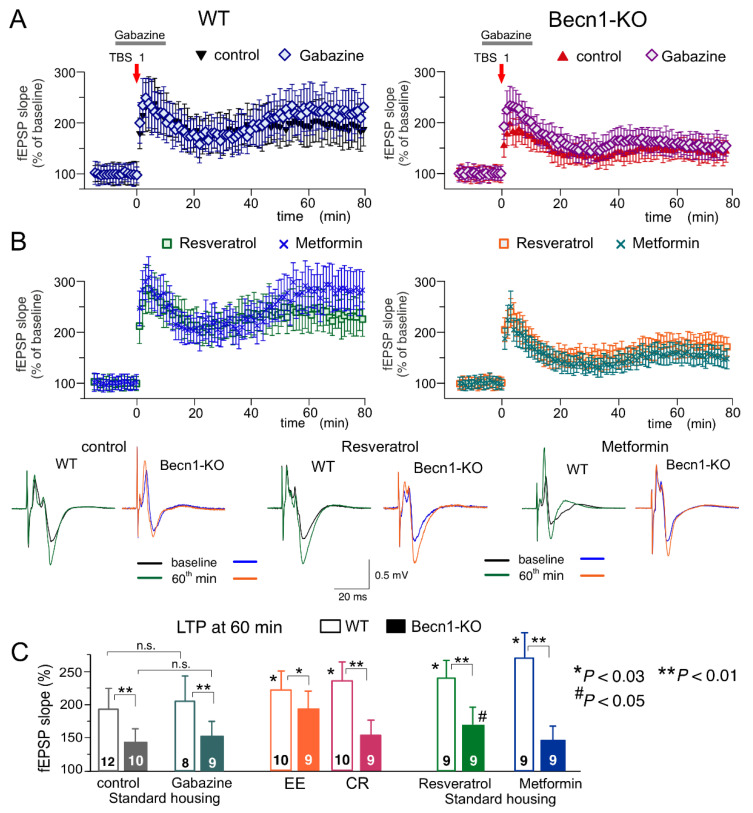
Impact of autophagy impairment on long-term synaptic plasticity in the CA1 hippocampal area. LTP of the field EPSPs was induced in the CA1 hippocampal area by one episode of theta-burst stimulation (TBS), as described in *Methods*. (**A**,**B**) The time course of changes in the slopes of fEPSPs recorded in the wild-type (left) and Beclin1-deficient mice (right) under different conditions. Dots in the graphs represent the average of six consecutive fEPSPs; data are shown as means ± SDs for 7–15 experiments (as indicated in panel C). Data were normalized to the fEPSP slope averaged over the 10 min period prior to the TBS. (**A**) LTP was induced in the CA1 neurons of SH mice under control conditions and in the presence of the GABA_A_ receptor antagonist gabazine (1 µM) applied, as indicated in the graphs. (**B**) The CA1 LTP recoded in the slices of SH mice after preincubation of brain slices with metformin (5 µM) or resveratrol (5 µM) for >4 h. The insets show the representative average fEPSP waveforms recorded before and 60 min after the TBS. (**C**) The pooled data on the magnitude of the CA1 LTP in mice of both genotypes at different conditions. Graphs show the magnitude of LTP evaluated as a relative increase in the fEPSP slope at 60 min, averaged across a 10 min time window. Data are shown as the means ± SDs for the numbers of experiments indicated. The statistical significance (two-population unpaired *t*-test) of the differences between genotypes and treatment groups is indicated by different symbols as follows: (*) effect of EE, CR or drug in the WT mice as compared to the control; (#) effect of EE, CR or drug in the Beclin1-deifient mice as compared to the control; (** or *) difference between WT and Beclin1-deficient mice. Note the significant decrease in the LTP magnitude in the Beclin1-KO mice and the significant effect of EE and resveratrol in the Beclin1-KO mice.

## Data Availability

The data presented in this study are openly available in FigShare repository at doi:10.6084/m9.figshare.20496516.

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
