# Peer review of "Impact of Autophagy Impairment on Experience- and Diet-Related Synaptic Plasticity"

_ijms, 2022, doi:10.3390/ijms23169228_

Round 1
Reviewer 1 Report
Lalo and coworkers examine glutamatergic mEPSCs, GABAergic mIPSCs and long-term potentiation of field EPSP in neocortical layer II/III neurons (Figs. 3-7) and long-term potentiation of field EPSP in hippocampal CA1 neurons (Fig. 8) in Beclin1-deficient mice to reveal the impact of autophagy on synaptic transmission and synaptic plasticity. The experiment was performed by using electrophysiological technique and multi-photon fluorescent imaging. As a result, they found in neocortical layer II/III neurons a glutamatergic transmission down-regulation, GABAergic transmission up-regulation, long-term potentiation impairment and a reduction in the effects of environmental enrichment (EE), caloric restriction (CR) and its pharmacological mimetic (metformin and resveratrol application) on synaptic transmission and plasticity. A similar long-term potentiation impairment and reduction in the effects of EE, CR and its pharmacological mimetic on synaptic plasticity were observed in hippocampal CA1 neurons. It was suggested that autophagy plays an important role in regulating excitatory, inhibitory transmission and long-term synaptic plasticity. This manuscript does not appear to be written carefully and there are many mistakes throughout the text. There are so many points that should be addressed and may be useful to amend this manuscript, as follows:
Major points:
1. Abstract: the authors do not give data about synaptic transmission in hippocampal neurons. From this point of view, this abstract should be rewritten.
2. Materials & Methods: there is no drug section. The authors should write what companies the drugs used were purchased from. Furthermore, please write the name of the place where the company such as Axon Instruments or WPI is located.
3. Figures: there is no explanation about the scale bar in Fig. 1A. Please amend this point. “Ina” in Fig. 1C should be “INa”. There is no scale bar in Fig. 1B and Fig. 2A. The time and current or voltage scale bar in Figs. 3A, 4A, 5A, 6A, 7A and 8B should be written more definitely.
4. Fig. 1: the data in Fig. 1B and C appear to be obtained from neocortical neurons, according to their legends. Where are corresponding data obtained from hippocampal CA1 neurons? Only Fig. 1A? Where are 2-photon fluorescent image data obtained from neocortical area? Please reply to these questions.
5. Fig. 2: it is not clear which data are obtained from Crx L2/3 or CA1. Please make this point clear.
6. Lines 246 and 247: the authors should state why the concentrations of metformin and resveratrol were used.
7. Figs. 3 and 4: both of mIPSCs and mEPSCs are recorded at -80 mV while there appears to be not so large differences in amplitude between mIPSCs and mEPSCs. What values are reversal potentials for mIPSCs and mEPSCs? Please reply to this question.
8. Lines 275 and 276: addition of TTX should be stated here. “of” should be put before “antagonists”. Please amend these points.
9. Line 342: what value was threshold of detection for mEPSC or mIPSC? Please reply to this question.
10. Lines 371 and 372: what are stable high-quality whole-cell recordings during 1 hour? This stability should be stated more clearly in terms of amplitude and frequency.
11. Line 503: what about the effect of gabazine on GABAergic synaptic currents in neocortical neurons? Was long-term potentiation in neocortical neurons not examined in the presence of gabazine? Please reply to these questions.
12. Lines 558 and 559: this sentence should be revised, because the authors do not give data about synaptic transmission in hippocampal neurons.
13. Discussion: the authors estimate synaptic transmission by observing “miniature” but not “electrically-evoked” synaptic transmission. In fact, synaptic transmission in the central nervous system is “evoked” but not “miniature” synaptic transmission. This point should be taken into consideration when discussing presynaptic effect.
Specific points:
1. Line 20: there are no explanations about metformin and resveratrol. Please amend this point.
2. Lines 47, 48, 62 and 63: “AMPK” and “mTOR” are defined twice. Please amend this point.
3. Line 113: is “Accordingly to” OK? Please check English.
4. Line 124: please use either “neurone” or “neuron” through the text. Where are data obtained from astrocytes? Please amend this point.
5. Line 131: please give values of the liquid junction potential compensated.
6. Line 170: is “accordingly” OK? Please check English.
7. Lines 207 and 211: neocortical pyramidal neurons are not located in the CA1 hippocampal area. Please resolve this contradiction.
8. Lines 230 and 231: are the data shown in Fig. 1B obtained from hippocampal CA1 neurons? Please reply to this question.
9. Line 251: pyramidal neurons of somatosensory cortex are not located in CA1 shown in Fig. 2C and D. Please resolve this contradiction.
10. Line 270: “A” in “GABAA” should be subscript throughout the text.
11. Line 280: “quantal size” should be explained shortly.
12. Line 288: “EE” is defined repeatedly (see line 28). In some places, both “EE” and “environmental enrichment” are used. The same is true for CE, which makes it difficult to read. Please amend this point throughout the text.
13. Line 305: there is no explanation about picrotoxin. Please amend this point. On the other hand, gabazine is also used (see line 502). Please write the reason why both of them were used. Addition of TTX should be stated here.
14. Line 308: is “Fig. 3” OK? Please amend this point.
15. Lines 323 and 352: it does not seem to be written as to how many events (sIPSCs or sEPSCs) the averaged record came from. Please amend this point.
16. Line 429: not “mice different” but “mice was different”.
17. Line 455: is “could counter-balance have” OK? Please check English.
18. Line 529: is “effect CR” OK? Please correct English.
19. Line 589: “dieses” should be “diseases”.
20. Line 601: is “Beclin1-deficient was” OK? Please check English.
21. Line 608: is “notion autophagy” OK? Please check English.
22. Line 612: is “degradation synaptic” OK? Please check English.
23. Line 646: is “degradation many” OK? Please check English.
24. Line 665: “Ca-signalling” should be “Ca2+-signalling”.
25. Lines 741-980: is it necessary to give all of the author names in ref. 32? Please reply to this question.
26. References: each letter in the title of ref. 14 begins with a capital letter, as different from that of ref. 15. References should be presented in a unified format. Moreover, please check all of the references whether they are cited correctly.
27. There appear to be more mistakes than pointed out above. Please check the manuscript very carefully.
Author Response
Major points
- We present pilot data on baseline synaptic transmission in the CA1 neurons in revised manuscript (extended Fig.3 and 4). The abstract has been clarified accordingly.
- The requested details have been included.
- All have been amended. In the Figures 5-8, all representative current/voltage waveforms are shown in the same scale, so the scale bars were omitted in some graphs to decrease the clattering (now this was clarified in the figure legends).
- The images and current waveforms have been shown for illustrative purposes, the examples would look rather similar for neocortical and hippocampal neurons. The examples of CA1 neurons were not shown to avoid clattering of the Figure. In the revised Figure 1, we included the representative images of CA1 area immunostaining in the panel A and added diagrams on functional readouts of CA1 neurons in the panels B and C.
- The Figure 2 was extended, and its legend was clarified.
- The concentrations of metformin and resveratrol were chosen based on literature data on their modulatory actions on autophagy and their blood plasma levels achieved in the pre-clinical studies in human patients; this is now mentioned in the revised manuscript.
- The reversal potential for EPSCs and IPSCs were about 0.5 mV and -21.5 mV correspondingly
- All amended as requested
- The threshold was for 2.5-2.8 pA; now it is mentioned
- Actually, all whole-cell recordings shown in the paper met the high-quality criteria (low access resistance, high input resistance, high signal/noise ratio and lack of large fluctuations of holding currents), so we omitted the “high-quality” from this sentence. The stability criteria were: variation in the access and input resistance and average mEPSC amplitude and frequency less than 15%, now it is mentioned both in the Methods and Results (sections 4.2 and 2.4).
- The action of gabazine in the neocortex was the same, i.e. moderate attenuation of synaptic currents and inability to rescue the LTP in Becn1-KO mice. Now it is mentioned in the revised manuscript.
- Now we present data on changes in synaptic transmission in CA1 neurons.
- In general, we agree with this point. Yet, we would like to emphasize that it is widely accepted that spontaneous miniature synaptic responses provide an adequate readout of the post-synaptic “side” of synaptic efficacy, which was the main aim of this paper. As for the presynaptic changes, we agree again with Reviewer, that analysis of frequency of miniature currents have limited capability in this regard. Still, the mEPSC frequency is widely used as main (and only) “pre-synaptic” parameter, even in high-impact research.
We have taken this into account and amended the Discussion.
Surely, changes in the evoked synaptic currents in the autophagy-deficient mice definitely deserve exploring and we are carrying out pilot experiments in our model.
Specific points:
1 - 3 All has been amended
- Astrocytes were mentioned by mistake, all was amended.
- The liquid junction potential was about 10 – 20 mV. One should note that the value of liquid junction potential is just a technical parameter of whole-cell voltage-clamp recordings, which has no biological meaning at all. It depends on the size and shape of micropipettes used and composition of intracellular and extracellular media. According to standard routine of voltage-clamp recordings, it is compensated prior to recordings by applying corresponding voltage to one input of differential amplifier, to obtain a zero value of holding current. Many makes of voltage-clamp amplifiers (like one we use) perform this routine automatically.
6 – has been corrected
7,8 the Figure 1 and its legend has been extended and clarified
- The legend for the Figure 2 was clarified.
- has been checked and corrected
- the term “quantal size” has been replaced with “quantal or unitary amplitude” which is easier to understand for non-specialists in synaptic physiology
- has been checked throughout the manuscript and corrected
- The picrotoxin, which is a non-competitive irreversible GABAA blocker, was used for pharmacological isolation of excitatory synaptic currents (this is now mentioned); this is routine practice in synaptic physiology. The gabazine (or SR 95531) is competitive, reversible antagonists (i.e. its action is much easier to titrate) was used for moderate, controllable attenuation of GABAergic signalling in the LTP experiments (no picrotoxin was used there).
14-24: all was amended
- The number of authors was truncated
- All checked and corrected
- The grammar and language have been thoroughly double-checked and corrected.
Reviewer 2 Report
The research manuscript of Ulyana Lalo and co-authors report the Impact of autophagy impairment on the experience- and diet-related synaptic plasticity.
The authors have addressed and explored the impact of autophagy on synaptic transmission and homeostatic and acute synaptic plasticity in neocortical and hippocampal neurons using transgenic mice with inducible deletion of Beclin1 protein. The underlying mechanisms involve the downregulation of glutamatergic and up-regulation of GABAergic synaptic currents and impairment of long-term plasticity in the Beclin1-deficient mice. The Beclin1-deficiency also significantly reduced the effects of environmental enrichment, caloric restriction and its pharmacological mimetics on synaptic transmission and plasticity.
In general, this study is partially novel and cannot provide potential evidence for the direct impact of autophagy on synaptic transmission and homeostatic and acute synaptic plasticity. Therefore, although there is not much research article so far describing about the autophagy on synaptic transmission. There are certain weaknesses in the experimental design of the molecular mechanism the author needs to look into it, to make the manuscript publishable form.
Major and Minor concerns:
1. In introduction, in final paragraph the authors need to specify their study aims clearly which can accomplish their hypothesis, be precise and clear about your objective.
2. In materials and methods, some of the protocols does not show the full details of the experiments, the authors need to look into it namely, immunohistochemistry, calcium experiment and how many animals were used in each group, N=?.
3. In methods for electrophysiology recordings, the protocol for LTP is not clear, please give the protocol in detail, what was the thickness of the brain slice, how did you cut the slice, whether you used artificial CSF, etc you need to clearly update the protocol for the readers to understand.
4. In figure 1A, the authors have shown pyramidal neurons staining, can they show the whole hippocampus of the brain slice.
5. In figure 1, authors need to check the figure legends, many spelling errors.
6. In figure 2, the authors need to include autolysosome experiment to show autophagosome and lysosome fusion and validate the autophagy induction using TEM imaging.
7. The authors need to check the figure legends of figure 3 and 4, it looks different font and there are some spell errors, kindly check.
9. A final schematic diagram would be better for the understanding of mechanism. The above corrections should be carried out in the figures and the requested experiments.
10. A careful English check and grammatical errors need to be resolved by the authors.
So, this manuscript precedes about the potential evidence for the direct impact of autophagy on synaptic transmission and homeostatic and acute synaptic plasticity. The underlying mechanism makes this work noble and creates scientific interest for the readers. However, the available research information seems to be insufficient for being accepted in current form. Taking together to all this issue I recommend minor revision to the current form the manuscript.
Author Response
- The final paragraph was amended
- The relation between sample size (experimental units) and animals is mentioned in the Methods, section 4.5. The samples size was indicated in the text and Figures or legends, where appropriate.
- We used the slices for the LTP experiments, as for whole-cell recordings (now it is clarified in the Methods). All necessary details of LTP experiments are given in the Methods (sections 4.1, 4.2) and Results (section 2.5).
- The representative images for CA1 staining have been added to the Figure 1.
5 and 7. The figure legends have been checked and corrected.
- The main aim of the experiments described in the Figure 2 was to verify the impairment of autophagy in the Beclin1-deficient mice. For this purpose, we used the BafylomicinA1-based assessment of autophagic flux is a conventional and widely recognised approach. Also, impairment of autophagy and decrease of autophagosomes and autolysosomes have been previously shown in neurons of Beclin-1 floxed mice (McKnight et al. ref [21] and Picford et al. https://www.jci.org/articles/view/33585). Our results showing deficit in autophagic flux in Beclin-1 floxed mice verify McNight et al. and Pickford et al. studies. Therefore, we strongly believe that no further experiments are necessary, bearing in mind rather limited time given for the revision.
- We think that it is somewhat prematurely to show the final schematic diagram since our study raised some questions which are yet to be resolved, in particular on molecular cascades underlying impact of beclin1 on excitatory synaptic signalling at pre-synaptic and post-synaptic loci. A good place for the final diagram might be a follow-up paper answering these questions or related review. Also, the revised paper is already rather extensive and contains 9 multi-panel figures.
All the corrections requested above have been made. - The grammar and language have been thoroughly double-checked and corrected.
Round 2
Reviewer 1 Report
Although this revised manuscript seems to be amended according to my comments, there remain to be still a lot of points that should be amended, as mentioned below. The authors do not seem to check this revised manuscript before submission.
1. Line 40: “caloric restriction” should be “CR” (see line 34). As both of them are used throughout the text, this is a little confusing. Please amend this point.
2. Line 87: this title should be amended, because Section 2.1 also gives data obtained from hippocampal neurons.
3. Line 96: not “Figure” but “Fig.”.
4. Fig. 1 and this legend: the preparation of data shown in the bar graph of Fig. 1C should be stated. Lines 104 and 105: not “Neun” but “NeuN”. Line 107: not “the cytosolic” but “The cytosolic”? (see line 103); please use uppercase or lowercase throughout figure legends. Line 118: Bar graph in Fig. 1C: not “INA” but “INa”. RIN and INa should be defined in this legend. Lines 117 and 118: not “amplitude Na” but “amplitude of Na”.
5. Line 124: not “Neun” but “NeuN”.
6. Line 131: since only INa is examined, “transmembrane currents” should be revised.
7. Fig. 2: “CA1” should be written in Fig. 2D as shown in Fig. 2C.
8. Line 171: “GABA” should be “GABAA”.
9. Line 201: please expand TTX and explain this action.
10. Lines 236 and 587: “antagonist” should be “receptor antagonist”.
11. Line 290: section 2.4 gives data from only neocortex neurons. Therefore, “neocortex neurons” should be added to this subtitle. Were the effects of autophagy modulators not investigated in hippocampal neurons?
12. Line 315: not “The” but “There”?
13. Line 319: please use either “hr”, “hour” (see line 294) or “h” (see line 329) throughout the text.
14. Lines 400, 425, 446 and so on: please use “LTP” here.
15. Lines 453 and 488: A in GABAA should be subscript.
16. Line 482: please correct the position of “(LTP)”.
17. Line 494: is “60th min” OK?
18. Line 507: not “have” but “has”?
19. Line 520: since the authors examine only input resistance and the peak amplitude of voltage-gated Na+ channel current, “basic neuronal functions” should not be stated here.
20. Line 684: it is not necessary to define LTP repeatedly (see line 386).
21. References: the authors do not seem to check them. Each letter in the title of ref. 27 begins with a capital letter, as different from that of ref. 28. References should be presented in a unified format. Is “Frontiers in cellular neuroscience” OK? (see “Front Mol Neurosci” in ref. 32) Moreover, please check all of the references whether they are cited correctly.
22. There appear to be more mistakes than pointed out above. If the authors prepare a re-revised manuscript, this should be checked very carefully before submission.
Author Response
We are very thankful to the Reviewer for very careful reading of our manuscript and constructive comments in both revision sessions. We did our best to correct all small errors and typos.
1-5: All was amended
- Was replaced for “Voltage-gated Na-currents”
7,8 Amended
- “TTX” was expanded; TTX was used to block putative firing of spontaneous action potentials in the presynaptic terminals, to ensure that minitiature spontaneous synaptic currents are elicited by release of individual vesicles rather than multi-vesicular package. This is a fairly standard approach to measure mEPSCs or mIPSCs; researchers working in the areas of Physiology or Neuroscience, for instance, become aware of this approach during their undergraduate years in lectures on vesicular synaptic transmission. So, we do not think that extensive explanation is really necessary, also, it would have clattered the Results section.
11 – We have preliminary data on the effects of autophagy modulators in CA neurons, both after dietary and in situ administration. They are in general agreement with data obtained in the neocortex and with data on changes in LTP; still the research is ongoing and we decided not to include the CA1 data in the current paper.
9-17 - Corrected
18 – “consequences” should have been here; corrected
19 – for clarity, we changed this for “basic electrophysiological characteristics “
20 – corrected
21 – The references have been managed by the referencing software embedded in the MSWord (EndNote), which follows format used by the specific journal (e.g. most of CellPress journals capitalize first letters in each word of title). Now we did reference list manually and double-checked it.
- The manuscript have been thoroughly double-checked and corrected